# Impact of COVID-19 on Sedation Requirements during Veno-Venous Extracorporeal Membrane Oxygenation for Acute Respiratory Distress Syndrome

**DOI:** 10.3390/jcm12103515

**Published:** 2023-05-17

**Authors:** Maria Paparoupa, Marlene Fischer, Hans O. Pinnschmidt, Jörn Grensemann, Kevin Roedl, Stefan Kluge, Dominik Jarczak

**Affiliations:** 1Department of Intensive Care Medicine, University Medical Center Hamburg-Eppendorf, Martinistr. 52, D-20246 Hamburg, Germany; 2Institute of Medical Biometry and Epidemiology, University Medical Center Hamburg-Eppendorf, Martinistr. 52, D-20246 Hamburg, Germany

**Keywords:** COVID-19 ARDS, analgosedation, sufentanil, propofol, midazolam, esketamine, volatile anesthetics, clonidine, dexmedetomidine, veno-venous extracorporeal membrane oxygenation

## Abstract

COVID-19-associated ARDS (C-ARDS) is mentioned to express higher analgosedation needs, in comparison to ARDS of other etiologies. The objective of this monocentric retrospective cohort study was to compare the analgosedation needs between C-ARDS and non-COVID-19 ARDS (non-C-ARDS) on veno-venous extracorporeal membrane oxygenation (VV-ECMO). Data were collected from the electronic medical records of all adult patients treated with C-ARDS in our Department of Intensive Care Medicine between March 2020 and April 2022. The control group included patients treated with non-C-ARDS between the years 2009 and 2020. A sedation sum score was created in order to describe the overall analgosedation needs. A total of 115 (31.5%) patients with C-ARDS and 250 (68.5%) with non-C-ARDS requiring VV-ECMO therapy were included in the study. The sedation sum score was significantly higher in the C-ARDS group (*p* < 0.001). COVID-19 was significantly associated with analgosedation in the univariable analysis. By contrast, the multivariable model did not show a significant association between COVID-19 and the sum score. The year of VV-ECMO support, BMI, SAPS II and prone positioning were significantly associated with sedation needs. The potential impact of COVID-19 remains unclear, and further studies are warranted in order to evaluate specific disease characteristics linked with analgesia and sedation.

## 1. Introduction

Originating from Wuhan, China, the severe acute respiratory syndrome coronavirus 2 (SARS-CoV-2) was identified to be the pathogenic agent causing coronavirus disease 2019 (COVID-19) [1]. SARS-CoV-2 primarily affects the respiratory system and causes viral pneumonia, which may lead to the development of acute respiratory distress syndrome (ARDS) with a need for invasive ventilation up to the necessity of extracorporeal membrane oxygenation (ECMO) [2,3,4]. The spread of the virus has caused a challenging pandemic for healthcare systems worldwide and more than 6.8 million confirmed fatalities so far [5]. At the beginning of the pandemic, intensivists worldwide observed that the majority of patients with COVID-19-associated ARDS (C-ARDS) required unusually high analgosedation in comparison to those with ARDS of other etiologies [6,7]. Several conjectures have been made about the pathophysiological background of this clinical phenomenon. Hanidziar et al. reported that high sedation requirements were likely related to the younger age and good state of health before the onset of C-ARDS. Moreover, high respiratory drive and intense inflammatory responses have been proposed as underlying mechanisms [8]. According to Kapp et al., the increased incidence of delirium and prolonged cognitive impairment may also have influenced the analgosedation needs in this patient cohort—or vice versa [9]. The seemingly higher requirement for sedation was suggested to become a point of pharmacological intervention, with the introduction of multimodal analgesia (gabapentinoids, intravenous lidocaine, esketamine) and early use of medications such as α-2 agonists, antipsychotics and sleep-promoting drugs such as benzodiazepines [10]. Compared with patients requiring fewer sedative agents, those who needed more were younger, had an increased body mass index (BMI) and had a lower PaO_2_/FiO_2_ (ratio of arterial partial pressure of oxygen to fraction of inspired oxygen) [11]. Especially severe ARDS treated with veno-venous ECMO (VV-ECMO) was linked to higher analgosedation, according to Flinspach et al. [12]. Controversial results have been published by Bohman et al., where sedation requirements did not significantly differ between patients with C-ARDS and non-COVID-19 related ARDS (non-C-ARDS) on VV-ECMO [13]. The objective of this monocentric retrospective cohort study was to compare the analgosedation needs between patients with severe C-ARDS and those with non-C-ARDS on VV-ECMO. 

## 2. Materials and Methods

### 2.1. Study Design and Study Population

This study was approved by the ethics committee at the Hamburg State Chamber of Physicians and conducted according to the Declaration of Helsinki (No. 2022-300239-WF). Owing to retrospective and de-identified data collection, the need for informed consent was waived. Demographic data, chronic comorbidities, cause of ARDS, VV-ECMO characteristics, sedation strategies and intensive care unit (ICU) outcomes were retrospectively collected from the electronic medical records of the patients and carefully reviewed by two intensivists. 

All adult patients (≥18 years old) with a polymerase chain reaction (PCR)-confirmed SARS-CoV-2 infection as the principal cause of severe ARDS with the need for VV-ECMO were included in the study (C-ARDS group). All patients were treated at our Department of Intensive Care Medicine between March 2020 and April 2022. The control group included patients with severe non-C-ARDS on VV-ECMO who were treated at the same department between the years 2009 and 2020. The sub-classification of non-C-ARDS included three categories: bacterial pneumonia, influenza-related pneumonia and other causes. Patients with veno-arterial ECMO or cardiac etiology of respiratory failure were excluded. The observational period started on the day of VV-ECMO implantation or on the day of admission to our department in case of cannulation at a referring hospital and simultaneous invasive mechanical ventilation via endotracheal tube. The observational period ended at the time of tracheotomy, explantation of VV-ECMO or death, depending on which event occurred first. Patients who were awake during VV-ECMO support, patients with tracheotomy before cannulation for VV-ECMO and patients with causes of respiratory failure other than ARDS were excluded from the study due to the different analgosedation regimes (Figure 1).

### 2.2. Management of VV-ECMO for Acute Respiratory Distress Syndrome (C-ARDS and Non-C-ARDS)

VV-ECMO was performed using Cardiohelp as well as Rotaflow systems (Getinge AB, Gothenburg, Sweden) and a Stöckert Centrifugal Pump & Console (PERFUSION.COM, Inc., Fort Myers, FL, USA). For cannulation, jugular and femoral veins were accessed and cannulas were placed percutaneously after vessel puncture with the Seldinger technique. This was performed by a cardiac surgeon or an experienced intensivist. Single patients with respiratory instability received cannulation for ECMO at the referring hospital and were transferred to our center with extracorporeal support. In these cases, cannulation and patient transport with ECMO were performed by one cardiac surgeon and one perfusionist. Cannula sizes (13–25 Fr) were chosen according to the vessel diameter as identified by ultrasound. Blood flow is adjusted to maintain arterial oxygen saturation ≥90% or arterial partial pressure of oxygen between 55 and 65 mmHg. The sweep gas flow was chosen to ensure the removal of carbon dioxide with a pH between 7.25 and 7.4. A reduction in arterial partial pressure of carbon dioxide of no more than 20 mmHg was aimed for at the start of VV-ECMO. 

During cannulation, a bolus of unfractionated heparin (5000 IE) was administered. The effect of heparin was measured with the activated clotting time during cannulation and the start of VV-ECMO. Thereafter, unfractionated heparin was administered continuously and monitored with the partial thromboplastin time (pTT). Target pTT was 40 to 50 s in all patients. Screening for heparin-induced thrombocytopenia was performed using the HIT 4T score [14]. In patients with a HIT score ≥ 4, an immunoassay was performed. In the case of laboratory-confirmed heparin-induced thrombocytopenia, argatroban was administered continuously for anticoagulation. Platelets were transfused in the presence of severe thrombocytopenia (<70,000 G/L). Fibrinogen levels were maintained above 1.5 g/L. 

### 2.3. Mechanical Ventilation Practices during VV-ECMO 

The management of mechanical ventilation was performed in accordance with the respective ELSO recommendations [15]. If possible, inspiratory plateau pressure was set below 25 cm H_2_O. A further reduction to <20 cm H_2_O was aimed for to avoid ventilation-related lung injuries. The setting of PEEP was chosen as ≥10 cm H_2_O to avoid atelectasis. Since both oxygenation and CO_2_ removal are primarily performed by VV-ECMO, respiratory rate was set to 4–30 breaths/min as well as FiO_2_ to 0.3–0.5 to avoid further lung damage.

### 2.4. Analgosedation Practice

The sedation strategy was primarily administered in analogy to the local sedation standard for critically ill patients with non-C-ARDS. The target Richmond Agitation–Sedation Scale (RASS) goal was set between −1 and 0. Thus, an escalation with further sedative agents was performed at the decision of the attending physician, in analogy to the in-house standards and comparable to the recommended ABCDE therapy bundle, which has been developed in parallel and verified in clinical studies [16]. Continuous intravenous (iv.) application of a strong opioid (sufentanil) in combination with continuous iv. application of sedatives was applied. Propofol was the first-choice sedative agent, followed by midazolam and/or esketamine, if bronchoconstriction was clinically relevant. In case of inadequate analgosedation, despite two or more systemic sedative agents, volatile anesthetics (isoflurane) were initiated. Clonidine or dexmedetomidine was used in the case of a prolonged necessity of sedation, in order to avoid propofol infusion syndrome.

Neuromuscular blocking was administered exclusively in case of uncontrollable patient–ventilator asynchrony. In order to exclude short-term deepening of sedation, e.g., bolus application for interventional procedures, we only considered regimes of more than four hours of continuous application for analysis. All patients received mechanical ventilation using an EVITA 500 (Dräger Evita Infinity V500) ICU ventilator, as well as multimodal therapy, according to the national and international guidelines for the C-ARDS treatment [17,18].

### 2.5. Data Analysis

Sedation requirements were categorized into quartiles for each agent, normalized to body weight and the duration of the observational period. The generated quartiles were scored as follows: 0 (substance not administered), 1 (low), 2 (moderate), 3 (high), 4 (very high) (Appendix A). With 7 different agents used for sedation and analgesia (sufentanil, propofol and/or midazolam and/or esketamine and/or clonidine and/or dexmedetomidine and/or isoflurane), each patient could maximally reach the score of 28 (all 7 agents, each in very high dosage).

We hypothesized that patients with C-ARDS required a higher amount of analgosedation during VV-ECMO, compared to ARDS attributable to other etiologies. The dependent variable was, therefore, the accumulative analgosedation during the study period, expressed in a sum score ranging from 1 to 28, which resulted from sedation dosage (dose/kg/time) divided into quartiles. The independent variable of primary interest was the COVID-19 infection, as it was considered to be the principal cause of C-ARDS (binary). The remaining independent variables with a potential influence on analgosedation needs were selected by clinical relevance: BMI (continuous), chronic comorbidities (categorical), chronic heart failure (CHF) (binary), acute kidney injury (AKI) on VV-ECMO requiring continuous renal replacement therapy (CRRT) (binary), the year of VV-ECMO (categorical), the performance of prone positioning during the VV-ECMO (binary) and the simplified acute physiology score (SAPS II) on admission (continuous).

All statistical analyses were conducted using SPSS version 29 (IBM Corp., Armonk, NY, USA). Baseline characteristics are presented as median and interquartile range (IQR) or absolute numbers and percentages. For group comparisons, *t*-tests, Mann–Whitney U tests, chi-square tests or Fisher’s exact tests were used as appropriate. To analyze the association between C-ARDS and analgosedation, we applied a general linear model with a negative binomial distribution and a log link function, using the SPSS generalized linear modeling tool GENLINMIXED, because it best described the data of the outcome variable, which were right-skewed and can be regarded as count data. C-ARDS, year of VV-ECMO and all secondary independent variables were included in the initial multivariable model as fixed effects. We then reduced the initial model by removing the non-significant secondary independent variables, following a stepwise backward approach. The final multivariable model thus only contained C-ARDS, year of VV-ECMO and secondary independent variables significant at the 0.05 level. Collinearity among independent variables was found to be of no concern, as judged based on their variance inflation factors and tolerance values.

## 3. Results

### 3.1. Descriptive Cohort Characteristics

A total of 365 patients could be included in this study. Of these, 115 (31.5%) were treated for C-ARDS and 250 (68.5%) for non-C-ARDS (Figure 1). Age, gender and SAPS II were comparable between both groups. Of the non-C-ARDS cases, 163 (65.2%) were attributable to bacterial pneumonia (community-acquired or nosocomial), 37 (14.8%) were due to influenza A and B pneumonia and 50 (20.0%) were related to other causes (aspiration, inhalation trauma, major injury, sepsis, intoxication, acute pancreatitis). The median BMI was 31.3 (IQR: 27.4–35.8) in the C-ARDS group versus 26.6 (IQR: 23.5–31.3) in the non-C-ARDS group. Detailed baseline characteristics as well as the prevalence of chronic underlying comorbidities of the study population are shown in Table 1.

Regarding the VV-ECMO characteristics (Table 2), prone positioning on VV-ECMO was significantly more frequent in the C-ARDS group (32.2% vs. 10.0%), and the requirement for CRRT under VV-ECMO was significantly more frequent in the non-C-ARDS group (61.6% vs. 38.3%). No patients in our cohort were extubated on VV-ECMO after cessation of sedation and spontaneous breathing trial. The mean ICU length of stay (LOS) was significantly higher in the C-ARDS group (35 vs. 24 days) but the overall ICU survival was similar between the two groups (40.9% for C-ARDS and 40.0% for non-C-ARDS population). The VV-ECMO characteristics of the study cohort are shown in Table 2.

### 3.2. Analgosedation Needs

Compared with the non-C-ARDS group, a higher proportion of patients with C-ARDS received sufentanil, midazolam, isoflurane, esketamine and dexmedetomidine. Instead, a higher dosage (dose/kg/time) of propofol was administrated in the non-C-ARDS group (Table 2, Figure 2). When comparing single substances, we found higher doses of sufentanil (*p* < 0.001) and midazolam (*p* < 0.001) in patients with C-ARDS compared with patients with non-C-ARDS. The sedation sum score differed significantly between the groups, with higher scores in C-ARDS compared to non-C-ARDS (*p* < 0.001) (Table 2). The quartiles for single substances are shown in Figure 3. COVID-19 was significantly associated with the requirement for sedation and analgesia in the univariable analysis (Table 3). However, the multivariable model did not show a significant association between COVID-19 and the overall sum score for sedation and analgesia, but the year of ECMO support, BMI, SAPS II and prone positioning were significantly associated with sedation needs (Table 3).

## 4. Discussion

The main findings of this study are as follows: (1) During VV-ECMO support, patients with C-ARDS require higher doses of analgesia and sedation compared with non-C-ARDS patients. (2) Multivariable generalized linear models did not show a significant association between C-ARDS and sedation requirement. (3) The year of ECMO support, BMI, disease severity and the use of prone positioning are associated with an increased requirement for analgesia and sedation. (4) We observed a distinct pattern of sedative and analgesic medication among the groups. 

Determining optimal sedation strategies remains a challenge in the treatment of ARDS and may be even more demanding in patients requiring ECMO [19,20]. Data on sedation requirements in patients with C-ARDS are conflicting. While Flinspach et al. found evidence for higher needs in C-ARDS during ECMO, the results from Sigala et al. suggest a differential pattern with a higher demand for dexmedetomidine but a reduced dosing requirement for propofol compared with non-C-ARDS [6,21]. Similarly, we found both higher doses of single substances, midazolam and sufentanil, and an overall increased demand for sedative agents in patients with C-ARDS compared with patients with non-C-ARDS. Importantly, we observed a distinct preference for anesthetic agents depending on ARDS etiology. A higher proportion of patients with C-ARDS received midazolam, isoflurane, esketamine and dexmedetomidine compared with patients with non-C-ARDS. In contrast, more patients with non-C-ARDS were administered propofol (Figure 2).

In line with Flinspach et al., we found a significant association between COVID-19 and the requirement for sedation and analgesia in univariable analysis. However, when adjusting for the year of ECMO support, the association became non-significant in the multivariable analysis due to a confounding effect. The lack of association in multivariable analysis may be attributable to several factors. First, we cannot rule out history bias affecting the findings of our study. Although there was no institutional change in sedation strategies in the years before the pandemic, we cannot retrospectively exclude clinical practice changing over time. The preference for anesthetic agents other than propofol may be attributable to the reduced availability of propofol during the COVID-19 pandemic. Despite reduced worldwide supply, propofol was available to a sufficient extent at our center. Yet, the threat of potential shortages may have influenced the preference for single substances. Second, we observed a significant association between patients who underwent prone positioning and sedation needs. Despite having been part of clinical routine management for moderate to severe ARDS for more than a decade [22], proning during VV-ECMO has become more widely used only in the past years [23,24,25]. Throughout the years 2021 and 2022, all ARDS patients receiving ECMO support at our center suffered from COVID-19-associated ARDS, whereas ARDS attributable to factors other than COVID-19 was not observed. As a result of prone positioning during VV-ECMO having become more widely used only recently and the almost exclusive etiology of COVID-19 causing severe ARDS with the necessity of VV-ECMO from 2020, proning was used 3 times more frequently for C-ARDS compared with non-C-ARDS. Third, we found a significant association between BMI and the requirement for sedation and analgesia. Findings from previous studies suggest increased sedation needs in patients with higher BMI [11]. In concordance with numerous reports, we observed a higher median BMI in patients with C-ARDS compared with patients with non-C-ARDS [26]. Therefore, the association between BMI and sedation requirement in our study population may be attributable to the higher prevalence of obesity in patients with COVID-19 [27]. Of note, both factors, the increased use of prone positioning in C-ARDS and the higher prevalence of obesity in C-ARDS, may have contributed to history bias and the lack of a statistical association between C-ARDS and sedation needs in our analysis. Fourth, increased inspiratory effort has been observed in patients with COVID-19, potentially contributing to patient self-inflicted lung injury, which may have similarly deleterious effects compared to those of ventilator-induced lung injury [28]. High inspiratory effort or patient–ventilator asynchrony may be encountered upon an increase in sedation depth to minimize the risk of additional lung injury [19]. It is important to note that we did not routinely measure the parameters of respiratory mechanics in the current study. Therefore, we can only speculate whether increased inspiratory effort in C-ARDS may explain the higher sedation needs compared with non-C-ARDS. 

### Limitations and Strengths

Strengths of this study include the large sample size which is higher than that from previously reported studies on sedation strategies in VV-ECMO for C-ARDS-associated respiratory failure. Furthermore, aside from comparing doses of single agents between patients with C-ARDS and non-C-ARDS, we here used a sum score for analgesia and sedation requirement that comprised quartiles of single substance dosing ranges, which allows for a more comprehensive assessment. Although arbitrary, the creation of this sum score allows for analyzing factors associated with an overall sedation need by multivariable analysis. The analysis of an overall, objective sedation score is one strength of this study and represents a novelty in study design compared with previous studies on sedation strategies in C-ARDS. 

Our study has several limitations that need to be addressed. Data for the control group with non-C-ARDS stem from electronic patient records that were collected over a 12-year period. Although sedation protocols for patients with VV-ECMO have not been edited throughout the study period, it seems plausible that changes in VV-ECMO management may have influenced our findings. For example, single agents such as dexmedetomidine have emerged, even if the intended target of sedation depth (as measured by the RASS) remained at 0 to −1. We tried to adjust for history bias by including the year of VV-ECMO treatment in the multivariable model. Yet, our results may be subject to history bias by the presence of unmeasured confounding variables, such as respiratory drive or changes in the education of physicians and the critical care team over the years. Importantly, the external validity of our findings is limited due to the single-center design. We, further, did not collect comparative data on respiratory drive or the frequency of patient–ventilator asynchrony. Future studies assessing requirements for sedation and analgesia in ARDS patients should collect data on inspiratory effort or surrogates of patient–ventilator asynchrony that may influence the dosing regimen. 

## 5. Conclusions

Patients with C-ARDS require higher doses of sedation and analgesia in comparison to patients with non-C-ARDS during VV-ECMO support. This effect may be attributable to the confounding effect of the year of ECMO support, BMI and an increased use of prone positioning during VV-ECMO treatment in C-ARDS. The potential impact of COVID-19 remains unclear, and further studies are warranted in order to evaluate specific disease characteristics linked with the requirement for analgesia and sedation.

## Figures and Tables

**Figure 1 jcm-12-03515-f001:**
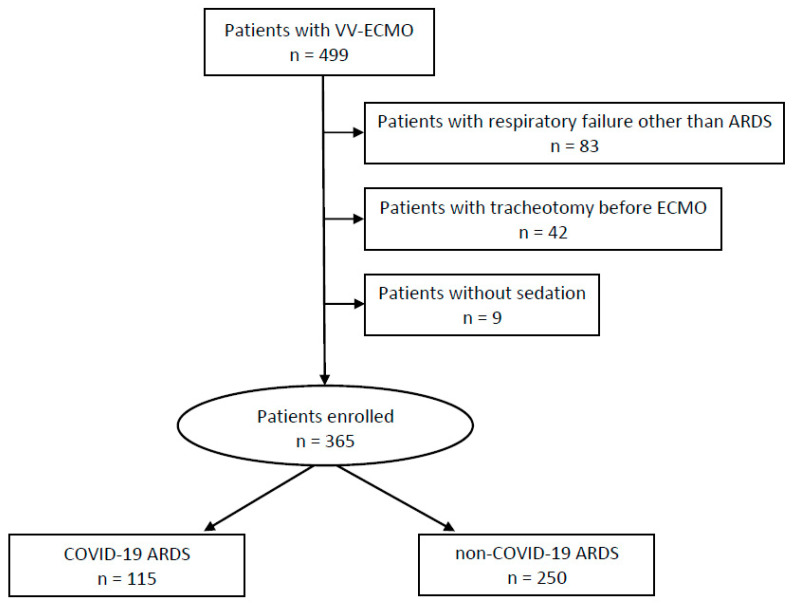
Study flow chart.

**Figure 2 jcm-12-03515-f002:**
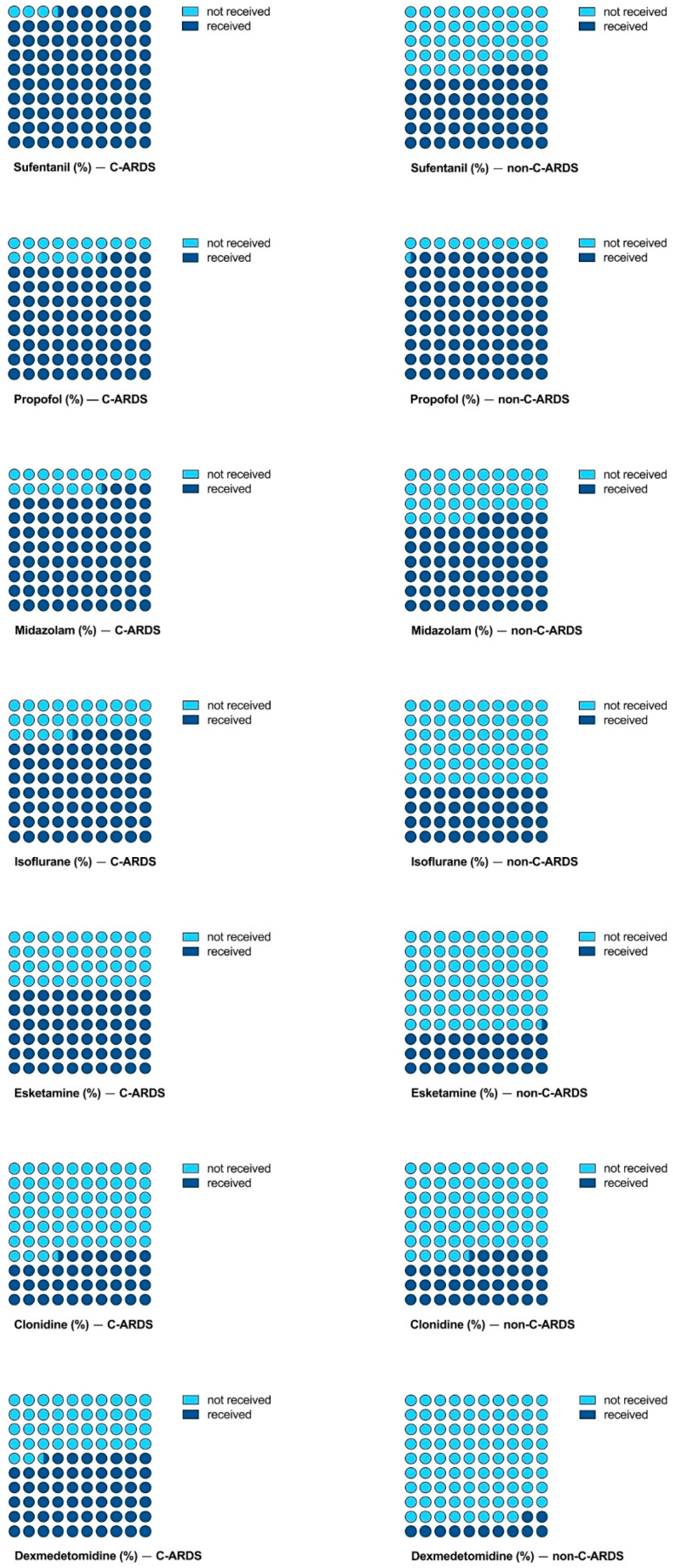
Proportion of patients who received single agents for sedation or analgesia, stratified by the etiology of ARDS.

**Figure 3 jcm-12-03515-f003:**
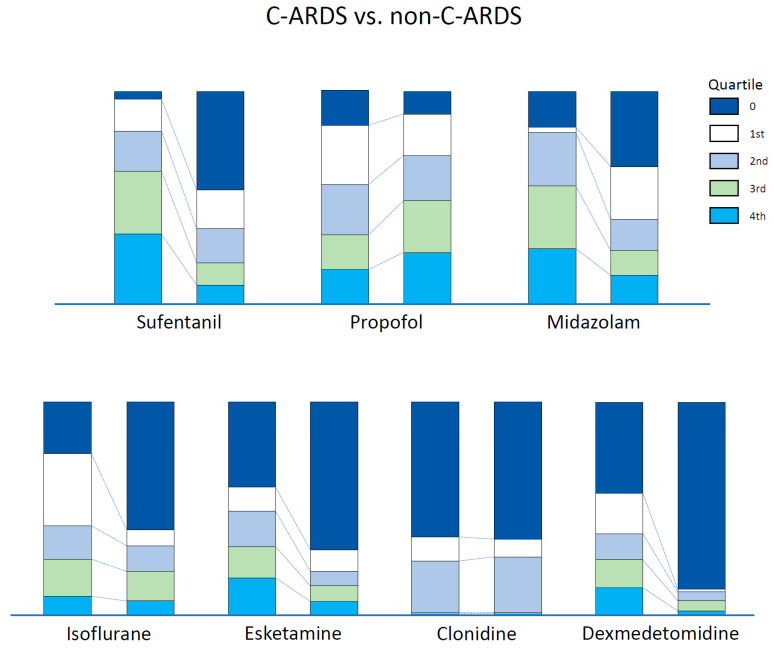
Dosing range of single substances, divided into quartiles with stratification by the etiology of ARDS.

**Table 1 jcm-12-03515-t001:** Baseline demographic characteristics.

Parameters	C-ARDS n = 115	Non-C-ARDS n = 250	*p*-Value
Demographics			
Age (years)	56 (48–62.8)	54 (44.3–63)	0.359
Sex			0.788
Male	78 (67.8%)	166 (66.4%)	
Female	37 (32.2%)	84 (33.6%)	
Height (cm)	176 (167.5–182)	176 (168–180)	0.934
Weight (kg)	95 (82–110)	82.5 (70–95.8)	<0.001
BMI (kg/m^2^)	31.3 (27.4–35.8)	26.6 (23.5–31.3)	<0.001
Comorbidities			
Arterial hypertension	55 (47.8%)	57 (22.8%)	<0.001
Chronic heart failure	5 (4.3%)	111 (44.4%)	<0.001
Diabetes mellitus	36 (31.3%)	48 (19.2%)	0.011
Chronic kidney disease (any stage)	4 (3.5%)	12 (4.8%)	0.784
Liver cirrhosis (any stage)	1 (0.9%)	0 (0%)	1.000
Chronic lung disease	17 (14.8%)	139 (55.6%)	<0.001
Immunosuppression	11 (9.6%)	68 (27.2%)	<0.001
Sum of comorbidities			0.003
0	34 (29.6%)	58 (23.2%)	
1	52 (45.2%)	84 (33.6%)	
2	20 (17.4%)	50 (20%)	
3	9 (7.8%)	45 (18.0%)	
4	0 (0%)	13 (5.2%)	
Reasons for VV-ECMO			---
COVID-19	115 (100%)	0 (0%)	
Bacterial pneumonia	0 (0%)	163 (65.2%)	
Influenza	0 (0%)	37 (14.8%)	
Other	0 (0%)	50 (20.0%)	
ICU characteristics			
SAPS II on admission	40 (33–48)	41 (34–49)	0.583
ICU survival	47 (40.9%)	100 (40.0%)	0.875
ICU length of stay (days)	35 (18–55)	24 (13–39)	<0.001

Data are expressed as n (%) or median (interquartile range). Abbreviations: BMI, body mass index; C-ARDS, COVID-19-associated ARDS; COVID-19, coronavirus disease 2019; ICU, intensive care unit; VV-ECMO, veno-venous extracorporeal membrane oxygenation.

**Table 2 jcm-12-03515-t002:** Therapy characteristics.

Parameters	C-ARDS n = 115	Non-C-ARDS n = 250	*p*-Value
ECMO therapy			
Time with ECMO and MV (hours)	442.0 (174.0–735.2)	170.1 (100.0–288.5)	<0.001
Days with ECMO (days)	19.7 (7.6–32.2)	10.1 (5.8–17.0)	<0.001
ECMO proning	37 (32.2%)	25 (10.0%)	<0.001
Tracheotomy	12 (12.2%)	124 (49.6%)	<0.001
CRRT	44 (38.3%)	154 (61.6%)	<0.001
Sedation			
Sufentanil (µg/kg/h ECMO)	0.58 (0.43–0.75)	0.41 (0.31–0.60)	<0.001
Propofol (mg/kg/h ECMO)	1.05 (0.26–1.92)	1.49 (0.67–2.20)	0.008
Midazolam (µg/kg/h ECMO)	42.22 (21.37–62.29)	8.60 (0.07–50.00)	<0.001
Isoflurane (ml/kg/h ECMO)	0.03 (0.02–0.05)	0.04 (0.03–0.05)	0.002
Esketamine (mg/kg/h ECMO)	0.51 (0.29–0.76)	0.45 (0.16–0.70)	0.139
Clonidine (µg/kg/h ECMO)	0.75 (0.41–1.12)	0.68 (0.51–0.92)	0.528
Dexmedetomidine (µg/kg/h ECMO)	0.45 (0.25–0.70)	0.58 (0.47–0.70)	0.061
Sum score of quartiles incl. sufentanil	12 (9–15)	7 (4–11)	<0.001
Year of ECMO			---
2009		3 (1.2%)	
2010		2 0.8%)	
2011		6 (2.4%)	
2012		3 (1.2%)	
2013		18 (7.2%)	
2014		24 (9.6%)	
2015		39 (11.6%)	
2016		51 (20.4%)	
2017		32 (12.8%)	
2018		29 (11.6%)	
2019		31 (12.4%)	
2020	42 (36.5%)	12 (4.8%)	
2021	62 (53.9%)	0 (0%)	
2022 (until April 30th)	11 (9.6%)	0 (0%)	

Data are expressed as n (%) or median (interquartile range). Abbreviations: C-ARDS, COVID-19-associated ARDS; COVID-19, coronavirus disease 2019; CRRT, continuous renal replacement therapy; ECMO, extracorporeal membrane oxygenation; MV, mechanical ventilation.

**Table 3 jcm-12-03515-t003:** Generalized linear model results for the association between clinically relevant variables and analgosedation needs.

Variable	B	95% CI	*p*
Univariable model			
COVID-19 (ref. non-COVID-19)	0.468	0.372, 0.563	<0.001
Multivariable model, first step			
COVID-19 (ref. non-COVID-19)	−0.135	−0.297, 0.026	0.099
BMI	−0.006	−0.010, −0.002	0.005
chronic comorbidities	−0.027	−0.080, 0.026	0.323
CHF	−0.070	−0.215, 0.074	0.337
AKI requiring CRRT	−0.034	−0.135, 0.068	0.514
year of VV-ECMO	0.125	0.096, 0.154	<0.001
prone positioning on VV-ECMO	0.122	0.016, 0.228	0.024
SAPS II on admission	−0.006	−0.010, −0.002	0.002
Multivariable model, final step			
COVID-19 (ref. non-COVID-19)	−0.054	−0.195, 0.086	0.445
BMI	−0.007	−0.011, −0.003	<0.001
year of VV-ECMO	0.119	0.090, 0.148	<0.001
prone positioning on VV-ECMO	0.119	0.013, 0.225	0.028
SAPS II on admission	−0.007	−0.010, −0.003	<0.001

Abbreviations: AKI, acute kidney injury; B, regression coefficient; BMI, body mass index; CHF, chronic heart failure; CI, confidence interval; COVID-19, coronavirus disease 2019; CRRT, continuous renal replacement therapy; *p*, *p*-value; SAPS II, simplified acute physiology score; VV-ECMO, veno-venous extracorporeal membrane oxygenation. Variables were eliminated stepwise backward. COVID-19 and the year of ECMO support were forced into the multivariable model.

## Data Availability

The datasets used and/or analyzed in the current study are available from the corresponding author upon reasonable request.

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
