# Peer review of "Impact of COVID-19 on Sedation Requirements during Veno-Venous Extracorporeal Membrane Oxygenation for Acute Respiratory Distress Syndrome"

_jcm, 2023, doi:10.3390/jcm12103515_

Round 1
Reviewer 1 Report
This is a very important topic. We all know that adequate analgesia and sedation are required after using VV-ECMO, especially in the severe stage of the disease. The performance of patients with COVID-19 is different from that of classic patients. Due to the violent reaction in the body, sedation and analgesia may reduce the respiratory drive, complete lung protection ventilation, and reduce the occurrence of lung injury. So is it possible to add some markers of inflammation and immunity? This is conducive to the accurate determination of the disease.
Reviewer 2 Report
Thank you for submitting this intersting study. Although your results seem clinically relevant and potentially useful, there are several issues that need to be addressed, before further consideration for publication.
Major, specific issues:
1. Analgosedation Practicet: In the control group, you included ARDS patients admitted to your ICU between 2009 and 2020. However, you state that sedatives escalation was performed according to the ABCDE bundle, which was published in 2014, according to your reference #14! Please clarify in detail!
2. Analgosedation and Methods in general: Please report your gas exchange targets and ventilation practices in the context of your ECMO protocol. Have these targets and practices remained the same throughout the 2009-2022 period? Also, please report the routine ECMO settings you use, as well as details about your ECMO equipment.
3. "Statistical analysis" should be changed to read "Data analysis". Also, please provide a detailed Table with your sedation score, including precise quartile cutoff values for each anesthetic agent so that the reader may understand what exactly you mean by low, moderate, high, and very high. Are quartile cutoff values "literature supported" or "arbitrary"?
4. Line 128: How exactly did you define the performance of prone positioning?
5. Multivariable analysis: In the discussion you mention that you actually used linear regression. I do not think that this analytic method is appropriate when you include binary and/or categorical variables in the model. I think that you should consider using generalized estimating equations (and repeat your analysis).
6. Discussion, Lines 217-220: Could differences in the use of specific anesthetic agents be primarily due to changes of sedation practices over time?
7. Discussion, lines 253-258: You did not actually assess respiratory drive in this study; therefore, this text seems too speculative; I would suggest adding to the limitations that there were no (comparative) data on respiratory drive or the frequency of patient-ventilator asynchrony.
8. Discussion, lines 267-269: you mention that your sedation score is "objective"; has this score been somehow been validated before its use in the current study? Again, which exactly are the "objective/non-arbitrary" criteria for the score's quartiles (and for each anesthetic drug)?
9. Limitations: please state examples of potential, unmeasured confounders.
10. Despite the several overall weaknesses of the paper, the conclusions seem to be supported by the reported results; however, again, I am concerned about the chosen linear analytic technique, despite the presence of binary and categorical variables.
Minor points:
1. Lines 84-87: Please edit to facilitate the reading of the text. For example, you may add "...patients with" before the word "tracheotomy" and replace "and" with "or" before the word "causes".
2. Results, descriptive characteristics: You should try to minimize repetition of the same results in the text and Table 1.
3. Line 160: Change "significantly higher" to "significantly more frequent".
Moderate editing is required.
Round 2
Reviewer 1 Report
It is suitable for publish in its current form.
Reviewer 2 Report
Thanks for the satisfactory revision. Please check carefully and confirm the accuracy of the numbers presented on the supplementary Table (especially for midazolam).
Minor English editing may still be needed.
